# Involvement of the SIX10 Gene in the Pathogenicity of *Fusarium oxysporum* Formae Speciales in Strawberries

**DOI:** 10.3390/ijms26031123

**Published:** 2025-01-28

**Authors:** Wenbo Yang, Tianling Ma, Dong Liang, Chuanqing Zhang

**Affiliations:** Department of Plant Pathology, Zhejiang Agriculture and Forest University, Hangzhou 311300, China; wbyang0811@163.com (W.Y.); liangdong0309@163.com (D.L.)

**Keywords:** *Fragaria* × *ananassa*, *Fusarium* wilt, *Fusarium oxysporum* f. sp. *fragariae*, secreted in xylem (SIX) gene, host specificity, pathogenicity

## Abstract

Strawberries are planted globally as an important crop. *Fusarium oxysporum* f. sp. *fragariae* (Fof), a haploid mitosporic, pathogenic fungus with obvious host specificity, is responsible for an economically devastating soil-borne disease seriously threatening strawberry. *Fusarium oxysporum* is distributed in soils worldwide and causes vascular wilt and root rot disease in over 100 plant species. However, the formae speciales of *F. oxysporum* commonly have a very narrow host range, often restricted to a single host plant species. We isolated and identified pathogenic *F. oxysporum* from diseased strawberry samples collected from different provinces in China. Further analysis showed that among the 55 *F. oxysporum* isolates, only 70.91% belonged to Fof, and the remaining 29.09% were named Fo. The mycelial growth of Fof was faster than that of Fo at 20, 30, and 35 °C. The sporulation ability of Fof was weaker than that of Fo, and Fof presented a significantly higher germination rate under high temperatures. Fof and Fo from strawberry were not pathogenic to tomato or cucumber plants, and Fof showed significantly higher pathogenicity on strawberry than Fo. To explore the pathogenic mechanism of Fof, we knocked out *SIX10* in Fof. The mycelial growth rate of *ΔFofSIX10* was significantly slower than that of the wild type, but there were no significant differences in spore production. The pathogenicity of *ΔFofSIX10* to strawberry was significantly weakened, showing decreased severity of symptoms, indicated by root and crown rot, and wilt. Our research provides a basis for understanding the interaction between *F. oxysporum* and the host strawberry and the occurrence and management *of Fusarium* disease on strawberry.

## 1. Introduction

Strawberry (*Fragaria* × *ananassa* Duch.) is a small berry, and China has the highest strawberry production and the largest cultivated area in the world [1]. It contains natural antioxidants, has a high nutritional and medicinal value, and plays an important role in protecting human health [2]. Regular consumption can effectively reduce the risk of cancer and heart disease [3]. However, *Fusarium* wilt is considered a destructive disease of strawberries [4], posing a serious threat to commercial strawberry production worldwide. This disease causes severe economic losses [5] and is very difficult to control [6]. Strawberry root rot is a major disease encountered in strawberry production [7], and it is considered the most common and destructive disease, leading to serious economic losses in the strawberry industry [8]. In recent years, crown rot has become a highly destructive disease affecting strawberry production in China [9].

The *Fusarium oxysporum* species complex (FOSC) comprises a group of diverse fungal pathogens capable of infecting over 100 plant species worldwide [10]. *Fusarium oxysporum* has specialized forms on different crops, such as *F. oxysporum* f. sp. *cubense* (Foc) on banana [11], *F. oxysporum* f. sp. *lycopersici* (Fol) on tomato [12], *F. oxysporum* f. sp. *melonis* (Fom) on melon [13], *F. oxysporum* f. sp. *crocus* on saffron [14], and *F. oxysporum* f. sp. *fragariae* (Fof) on strawberry [6,15,16,17]. Fof is a soil-borne pathogen with characteristics of the FOSC and is difficult to control chemically due to its easy mutability [18]. It is known as the most problematic factor in strawberry production [19]. It has been reported that Fof accounts for all isolates for pathogenic *F. oxysporum* on strawberry and is aggressive [20]. Therefore, it is important to study the pathogenic mechanism of Fof in strawberries to prevent and control strawberry diseases.

Long-term interactions between plants and pathogenic isolates have resulted in complex attack and defense mechanisms [21]. The demonstration of lateral transfer of the *F. oxysporum* gene provides more powerful proof for the study of genes related to key pathogens [22]. Compared with other *Fusarium* species, the pathogenicity of *F. oxysporum* varies among crops and varieties of the same crop, exhibiting a high degree of host specificity. The most in-depth research includes the *secreted in xylem 1* (*SIX1*) gene [23]. Tintor et al. [24] found that *SIX1* and *SIX4* of *F. oxysporum* isolate *Fo5176* acted in exosomes. Dilla-Ermita et al. [25] found that, although all *SIX6* knockout isolates were virulent on *FW1*-resistant strawberry varieties, these varieties were still less virulent than *fw1/FW1*-sensitive control varieties. Thatcher et al. [26] found that *F. oxysporum* f. sp. *conglutinans* required *SIX1* homologs for full virulence. Poon et al. [27] found that three *FocSIX* (*FocSIX1b*, *FocSIX1c,* and *FocSIX8a*) genes were only expressed after induction by the host; thus, they could be used as biomarkers to detect the early onset of banana wilt disease. Yang et al. [10] demonstrated that *FovSIX16* was essential for the complete virulence of *F. oxysporum* f. sp. *Vasinfectum7* (*Fov7*). However, there is currently a lack of well-known publications regarding *SIX10* in Fof. Therefore, to explore the relationship between *SIX10* and Fof, this study compared and analyzed Fof and other *F. oxysporum* isolates and found that *SIX10* affected the growth of Fof and reduced its pathogenicity.

## 2. Results

### 2.1. Isolation and Identification of Pathogenic Fusarium oxysporum

A total of 55 isolates were obtained from strawberry plant samples with wilt symptoms collected from Anhui, Guizhou, Hubei, Liaoning, Yunnan, Zhejiang, and Sichuan Provinces. After identification by the translation elongation factor (*EF-1α*) gene and internal transcribed spacer (ITS) region, these isolates were identified as *F. oxysporum*. A phylogenetic tree for *Fusarium* was constructed based on the *EF-1α* gene and ITS sequence, and it consisted of two clades (Figure 1). Clade-1 contained 21 Fof isolates, while clade-2 contained 18 Fof isolates.

The most Fo isolates were collected from Yunnan Province, with 18, followed by Zhejiang Province, with 17. Less than 10 isolates were obtained in the other five provinces. Among the Fo isolates obtained from strawberry samples collected from Anhui, Hubei, Yunnan, and Zhejiang Provinces, Fof accounted for over 80% (Table 1).

### 2.2. Fusarium oxysporum f. sp. fragariae Is Partially Responsible for the Pathogen Population

Genomic DNA was amplified from 55 *F. oxysporum* isolates using PCR with the FofraF/FofraR primer pair (Table 2). Agarose gel electrophoresis showed that 39 *F. oxysporum* (Fo) isolates were successfully amplified, showing 239 bp target bands. These were identified as *F. oxysporum* f. sp. *fragariae* (Fof), accounting for 70.91% of the isolates (Figure 2).

### 2.3. Growth, Sporulation, and Germination Abilities of Fo and Fof at Different Temperatures

Fof and Fo grew under five temperature conditions, with Fof showing stronger adaptability to high temperatures (Figure 3A). At 25 and 28 °C, there were no significant differences in the growth rate (*p* > 0.05). At 20 °C, the mycelial growth rate of Fof was significantly higher than that of Fo (*p* < 0.05). At 30 and 35 °C, the mycelial growth rate of Fof was significantly higher than that of Fo (*p* < 0.01).

Fof and Fo both produced spores under five temperature conditions, with Fo having significantly higher spore production than Fof (Figure 3B). The spore production of Fo was significantly different (*p* < 0.05) at 20, 25, 28, and 30 °C. The spore production of Fo was significantly higher than that of Fof (*p* < 0.05), but there was no significant difference in spore production between Fof and Fo at 35 °C (*p* > 0.05).

At 35 °C, Fof and Fo spores did not germinate. At 20 and 25 °C, there were no significant differences in the spore germination rate between Fof and Fo (*p* > 0.05). At 28 °C, the spore germination rate of Fof was significantly higher than that of Fo (*p* < 0.05). At 30 °C, the spore germination rate of Fof was significantly higher than that of Fo (*p* < 0.01) (Figure 3C).

The pathogenicity of Fof was significantly higher than that of Fo. Fo and Fof were not pathogenic to strawberry plants at 20 or 35 °C, but they had the strongest pathogenicity at 28 °C (Figure 3D).

### 2.4. Pathogenicity of Fo and Fof

The pathogenicity study showed that both Fof and s-Fo (*F. oxysporum* from strawberry) caused crown rot disease in strawberry, with the crown turning brown (Figure 4) and the plants being significantly dwarfed. The degree of dwarfism in strawberry plants inoculated with Fof (*F. oxysporum* f. sp. *fragariae*) was significantly higher than that in those inoculated with s-Fo. Fof and s-Fo were not pathogenic to tomato or cucumber plants. Fol (*F. oxysporum* f. sp. *lycopersici*) was only pathogenic to tomato, and Fom (*F. oxysporum* f. sp. *melonis*) was only pathogenic to cucumber. The symptoms were characterized by crown overflow and shrinkage, but Fol and Fom were not pathogenic to the strawberry crown (Figure 4).

### 2.5. SIX Homologous Gene Evolutionary Tree Analysis

In the gene evolutionary trees of the *SIX10* gene, *Fof16180* was clustered in the same branch as other seven formae speciales of *F. oxysporum*, suggesting that *Fof16180* is related to other formae speciales of *F. oxysporum* (Figure 5).

### 2.6. Deletion of Fof16180 in Fof

Three *Fof16180* gene deletion mutants (*ΔFof16180* is the equivalent of *ΔFofSIX10*) were obtained and identified using PCR analysis with the primer pair *Fof16180*-ID-F/*Fof16180*-ID-R. This primer pair amplified 1604 bp fragments from the *Fof16180* mutants (Figure 6B).

### 2.7. Growth, Spore Production, and Spore Germination of ΔFof16180

On potato dextrose agar (PDA) plates, there were no significant differences in colony morphology between *ΔFof16180* and wild-type H6. The middle of the mycelium was light purple, and the edges were white. The growth rate of *ΔFof16180* was significantly lower than that of wild-type H6 on PDA plates (Figure 7A).

There were no significant differences in spore production or germination rate between *ΔFof16180* and the wild type from 6 to 14 h (Figure 8).

### 2.8. Effect of Abiotic Stress on ΔFof16180

*ΔFof16180* grew faster under the stress of Congo red, KCl, and NaCl, and its growth was significantly higher than that of the wild type. *ΔFof16180* grew slower under SDS stress, and its growth was significantly lower than that of the wild type. *ΔFof16180* grew faster under glucose stress, and there were no significant differences between *ΔFof16180* and the wild type (Figure 9).

### 2.9. Effect of ΔFof16180 on Pathogenicity

After root inoculation, *ΔFof16180* significantly reduced the pathogenicity of wild-type isolate H6 on plants, but the pathogenicity of *ΔFof16180* on roots and crowns was not different from the wild-type isolate (Figure 10). The disease index of *ΔFof16180* in strawberry plants and roots was reduced compared with wild-type H6, but there were no changes in the disease index at the crown (Table 3).

Crown inoculation showed that *ΔFof16180* significantly reduced the pathogenicity of wild-type isolate H6 on the plants, roots, and crowns (Figure 11). The disease index of *ΔFof16180* was reduced in strawberry plants, roots, and crowns compared with the wild-type H6 (Table 3).

In general, *ΔFof16180* reduced the symptoms of plant wilt.

## 3. Discussion

FOSC is a group of diverse filamentous, ascomycetous fungi that are soil-borne and cause disease in many economically important crop species [28,29,30]. *Fusarium oxysporum* f. sp. *fragariae* has been found in Australia, Japan, California, Spain, China, and other places around the world since the 1960s [31,32,33,34]. In our study, we isolated pathogens from strawberry wilt, but not all pathogens isolated from strawberry with symptoms of wilt disease were *F. oxysporum* f. sp. *fragariae*; some were *F. oxysporum*. We tested the pathogenicity of both specialized and non-specialized isolates and found that *F. oxysporum* f. sp. *fragariae* had stronger pathogenicity in strawberry than other *F. oxysporum* specialization types. Moreover, the differences in growth rate between *F. oxysporum* and *F. oxysporum* f. sp. *fragariae* were even greater at high temperatures.

Most researchers believe that the factors that determine pathogenicity also endow pathogens with host specificity [35]. Studies have shown that the *SIX* gene is associated with host specificity to pathogens [36]. In our study, we knocked out the *SIX10* gene and obtained a deletion mutant, *ΔFof16180*. Routine biological characterization and pathogenicity testing were conducted on *ΔFof16180*. The mycelial growth rate of *ΔFof16180* was significantly lower than that of wild-type isolate H6 on PDA, but there were no significant differences in spore production between the deletion mutant and the wild type. At the same time, the growth rate of *ΔFof16180* differed on CM, MM, and OA media but was not significantly slower than that of the wild type. This indicates that the *SIX* gene is not necessary for fungal growth. Studies have shown that disrupting the *FonSIX6* gene in *Fon* race 1 did not affect its sporulation or growth rate [37]. The growth rate of *ΔFof16180* also differed due to different abiotic stressors. *ΔFof16180* was still pathogenic to strawberry plants, but compared with wild-type isolate H6, its overall pathogenicity was reduced, indicating that *SIX* genes play an important role in pathogenicity. Studies have shown that the *SIX1* and *SIX6* deletion mutants of Fol cause diseases [38]. Disrupting the *FonSIX6* gene in Fon race 1 significantly enhances Fon virulence in watermelon [37].

Why are other *F. oxysporum* specialization types not pathogenic to strawberries? Does the greater difference between *F. oxysporum* and *F. oxysporum* f. sp. *fragariae* at high temperatures mean that *F. oxysporum* f. sp. *fragariae* has stronger high-temperature resistance? Throughout the strawberry production period, the seedling stage is mostly under high-temperature conditions. Does this mean that there is a greater risk of *F. oxysporum* f. sp. *fragariae* in production? These issues are worth further research. The expression and function of *SIX* genes in the hyphal growth, sporulation, and pathogenicity of *F. oxysporum* f. sp. *fragariae* require further validation. Our findings provide new insights into the interaction between pathogenic microorganisms and host plants.

## 4. Materials and Methods

### 4.1. Media

Table 4 shows the preparation methods of the media.

### 4.2. Isolates, Molecular Identification, and Phylogenetic Analysis of F. oxysporum

The isolates used in this study were obtained from strawberry samples with wilt symptoms collected from Anhui, Guizhou, Hubei, Liaoning, Yunnan, Zhejiang, and Sichuan Provinces. The samples were washed with tap water, cut into small pieces with sterilized scissors, soaked in 1% sodium hypochlorite solution for 1 min, soaked in 70% alcohol for 1 min, washed with sterile distilled water three times, and dried with sterile filter paper [40]. Each tissue sheet was placed in a 90 mm Petri dish containing PDA medium supplemented with streptomycin sulfate (100 mg/L) and cultured in an incubator at 28 °C in the dark. After 3–5 d of incubation, the mycelia were transferred to a new PDA plate [41]. Single-conidium isolates were stored on PDA slants at 4 °C.

The total DNA of each tested isolate was extracted using a Fungi Genomic DNA Rapid Extraction Kit (B518229-0100; Sangon Biotech, Shanghai, China). The *EF-1α* gene and ITS region were amplified using the primer pairs EF-1/EF-2 [42] and ITS1/ITS4 [43] (Table 1). The sequenced and spliced sequences were retrieved from GenBank, the nucleic acid sequence database of the National Center of Biotechnology Information (NCBI, Bethesda, MD, USA), to select the standard isolates. The login number of the standard isolate was obtained through a literature search, and its sequence information was downloaded from GenBank. The sequences of the standard and isolated isolates were aligned and corrected in Fasta format in MEGA 7.0 software [44]. The corrected individual gene sequences were concatenated in Sequence Martix 1.8, and a sequence alignment file for tree building was generated [45]. Modeltest3.7win, Win-pup4b10 console, and Mrmodeltest2 were used to implement the optimal models for estimating nucleotide substitutions in MrMTgui [46]. Based on the *EF-1α* gene and ITS sequence, Bayesian inference phylogenies [47] were used to establish a phylogenetic tree. The establishment of an evolutionary tree for *SIX* genes was also based on this method.

### 4.3. Identification of F. oxysporum f. sp. fragariae

Fof-specific primers FofraF/FofraR (Table 1) were designed using Suga based on whole-genome alignment results [48]. First, the genomic DNA of Fo isolates was extracted using the method described in Section 4.2, and the genomic DNA of the 55 isolates was amplified using primer pair FofraF/FofraR. The total volume of the PCR system was 25 μL, with 1 μL of upstream and downstream primers, 1 μL of template, 12 μL of 2 × Taq PCR MasterMix, and ddH_2_O for a final volume of 25 μL. The PCR amplification program was as follows: pre-denaturation at 95 °C for 3 min, denaturation at 95 °C for 15 s, annealing at 63 °C for 30 s, extension at 72 °C for 15 s, and final extension at 72 °C for 10 min. The amplification product was electrophoresed in 1% TAE buffer using 1% agarose gel, and those with amplified 239 bp target bands were called Fof.

### 4.4. Growth, Spore Production, and Spore Germination of Pathogens

Mycelia plugs (5 mm in diameter) were transferred to new PDA plates and cultivated under dark conditions at 28 °C. After several days of cultivation, the size of the colony was measured. The mycelial growth and spore production rates were calculated after 7 d.

A spore suspension was obtained using the static spore production method [49] and cultured in a constant temperature incubator at 28 °C for 7 d with three replicates for each isolate. A 5 mL volume of double-distilled water (ddH_2_O) was added to each culture dish. The colony surface was gently scraped with a glass slide for spore overflow. The spore suspension was obtained by filtering with three layers of lens paper and shaken thoroughly, and 8 μL was aspirated using a pipette. The spores in suspension were counted using a hemocytometer, and each isolate was observed in at least five fields.

Using the spore germination method [50], the spore concentration was adjusted to 1 × 10^6^ spores/mL with ddH_2_O, and 20 μL was collected and placed onto WAM. It was spread evenly with a sterilized coating rod and placed in a constant-temperature incubator at 28 °C. Each isolate was set up with three replicates, and the number of germinated spores and the total number of spores were observed and recorded. The total number of spores in each field of view was not less than 100. The following formula was used to calculate the spore germination rate: Germination rate (%) = (number of spores germinated/total number of observed spores) × 100.

### 4.5. Pathogenicity Assays of Different Specialized Types of F. oxysporum

Six-week-old robust strawberry, tomato, and cucumber plants were selected for pathogenicity testing. The different *F. oxysporum* specialization types included *F. oxysporum* f. sp. *lycopersici* (Fol), *F. oxysporum* f. sp. *melonis* (Fom), *F. oxysporum* f. sp. *fragariae* (Fof), and *F. oxysporum* from strawberry (s-Fo). Spore suspensions were obtained using the static spore production method [49]. The spray inoculation method was used. Sterilized insect needles were used to stab the crowns of strawberry, tomato, and cucumber plants. Each plant was sprayed with 5 mL of a spore suspension with a concentration of 1 × 10^6^ spores/mL. Each isolate was used to inoculate five plants. After inoculation, plants were placed in a 28 °C, 12/12 h light/dark alternate incubator for moisturizing culture. The incidence of the disease was recorded 15 d later.

### 4.6. Construction of Fof16180 Deletion Mutants

The double-joint (DJ) PCR approach [51] was used to generate the gene replacement construct for *SIX10*. Briefly, two primer pairs (*Fof16180*-UP-F/*Fof16180*-UP-R and *Fof16180*-DOWN-F/*Fof16180*-DOWN-R) (Table 1) were used to amplify the upstream and downstream sequences of *Fof16180* based on the wild-type H6 genome. The primer pair HPH-F/HPH-R was used to amplify a 1349 bp fragment encoding the HPH cassette, containing the hygromycin-resistant gene and the trpC promoter. The three amplicons (upstream, HPH cassette, and downstream) were fused in a second round of DJ PCR. Based on the fused fragment, the final PCR products with an overlapping part of 3089 bp were amplified using the nested primer pair *Fof16180*-Nest-F/*Fof16180*-Nest-R (Table 1).

### 4.7. PEG-Mediated Fof Protoplast Transformation

In a sterile environment, the mycelium pieces at the edge of the colony were collected using a sterile toothpick, probed for breakage, and placed in 30 mL YEPD culture solution. Cultures were stored at 28 °C for 24 h at 180 rpm and observed during this period. A sterile steel ball was added to the 2.0 mL aseptic centrifuge tube, and a small fungus ball was obtained after shaking and placed in a centrifuge tube using a 1 mL pipette. It was not overfilled to prevent contamination. A multi-sample tissue grinder (45 Hz, 80 s) was used to break up the mycelia on the small fungal balls. The broken mycelia were transferred to YEPD according to the amount of mycelia required. Generally, 2 mL of broken mycelia was used to inoculate 100 mL of YEPD, and four tubes were transformed. Young mycelium was obtained by shaking at 180 rpm for 12–16 h at 28 °C. After this step, the growth state of mycelia was observed at any time after 12 h, and the aging of mycelia was avoided.

Three cell wall lyases (0.25–0.3 g of cellulase, 0.25–0.3 g of lysozyme, and 0.07–0.1 g of driselase from *basidiomycetes* sp.) were weighed and dissolved in 10 mL of 0.7 M NaCl solution. The enzyme solution was filtered with a 0.45 or 0.22 μm bacterial filter. The young mycelium was filtered with sterile three-layer lens paper, added to the sterile enzyme solution, and shaken well at 100 rpm for 3 h at 30 °C. The protoplasm cleavage was examined using a microscope. In a pre-cooled centrifuge, the enzymolysis mixture was filtered using sterile three-layer lens paper as the filter paper on ice. Enzymatic hydrolysis of the filtrate was performed with a sterile 2 mL centrifuge tube, avoiding light and low temperatures. The protoplasts were precipitated with 1 mL of 0.7 mol NaCl solution in the resuspension and centrifuged at 5000 rpm for 5 min at 4 °C. The supernatant was discarded, and the protoplasm in the enzymolysis mixture was enriched. The protoplast was precipitated by resuspending it in 1 mL of STC solution and centrifuging it at 5000 rpm for 5 min at 4 °C. The supernatant was discarded, and the protoplasm was left to precipitate.

The protoplasts were precipitated with 750 μL of STC solution and resuspended. Then, 100–200 μL of target DNA or plasmid solution (up to 200 μL), 5–10 μL of heparin sodium (depending on the protoplasm concentration), and 200 μL of SPTC (added drop by drop) were added, gently mixed, and left on ice for 30 min in the dark. SPTC of 1/3 volume of the above mixture was added drop by drop, and the solution was mixed gently and left for 20 min at room temperature in the dark. The transformation system was added to 20 mL of RM liquid medium and cultured overnight at room temperature (28 °C) for 15–30 min in the dark and shaken at 100 rpm. The next day, the recovered mycelia were mixed with SRM medium at a moderate temperature. Antibiotics were added, and the inverters were selected after being cultured on a plate at 28 °C for 2–4 d.

### 4.8. Effect of Abiotic Stress on Deletion Mutants

In non-biological stress experiments, Congo red, KCl, SDS, glucose, and NaCl were used for acid stress, salt stress, cell membrane stress, nutritional stress, and osmotic stress, respectively. The inoculation method was the same as that described in Section 4.4.

### 4.9. Pathogenicity Assays for Deletion Mutants

Pathogenic inoculation was divided into two parts: the root and the crown. Root inoculation was performed using the soaking method [50]. The roots of strawberry seedlings were immersed in a spore suspension with a concentration of 1 × 10^6^ spores/mL for 20 min, and sterile water was used as the blank control. The crown was inoculated using the method described in Section 4.5. After inoculation, they were placed in a light incubator to observe the disease incidence.

The disease incidence of the plant was determined according to the following grades: 1 = no disease symptoms; 2 = 25% of plant wilted or necrotic; 3 = 50% of the plant wilted or necrotic; 4 = 75% of the plant wilted or necrotic; and 5 = dead plant [52].

The disease incidence in the roots was determined according to the following grades: 0 = no disease symptoms; 1 = the main root is lightly brown, and the lateral root growth point is diseased, but the plant can grow normally; 2 = the main root is diseased and turns from brown to black, and the tip of the lateral root turns black, but the plant can still grow normally; 3 = most of the main root turns black and cannot continue to grow, the lateral roots are reduced or missing, and the plant grows slowly; and 4 = the roots are completely rotten, and the plant cannot continue to grow and may even wilt and die [53].

The disease incidence in the crown was determined according to the following grades: 0 = no rot present on the crown surface; 1 = 25% of the crown surface affected by rot; 2 = 50% of the crown surface affected by rot; 3 = 75% of the crown surface affected by rot; and 4 = entire crown surface affected by rot and extending to the finger stalk or pulp [54].

By observing the disease severity of the interaction between the pathogen and strawberry, based on the grading standards of the three parts mentioned above, the disease level was calculated as the disease index according to the following formula:

Disease index = ∑ (number of plants infected × their intensity grades)/(total number of plants observed) × (maximum severity grade) × 100 [53].

### 4.10. Statistical Analysis of Data

We used SPSS 29.0 and Prism 10 to analyze and process the statistics.

## Figures and Tables

**Figure 1 ijms-26-01123-f001:**
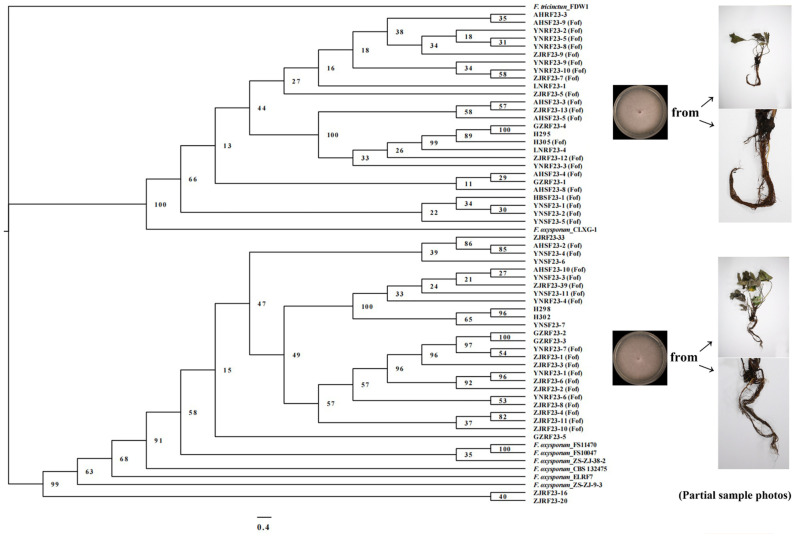
The Bayesian inference phylogeny method was used to construct a *Fusarium oxysporum* phylogenetic tree based on the *EF-1α* gene and ITS sequence. Dimensions: ntax = 63, nchar = 1477; format: data type = dna; outgroup = *F. tricinctum*; Lset: nst = 2, rates = invgamma; Prset: statefreqpr = dirichlet(1, 1, 1, 1); mcmcp: savebrlens = yes, ngen = 100,000, samplefreq = 100, nchains = 4; sumt: contype = allcompat, burnin = 5000.

**Figure 2 ijms-26-01123-f002:**
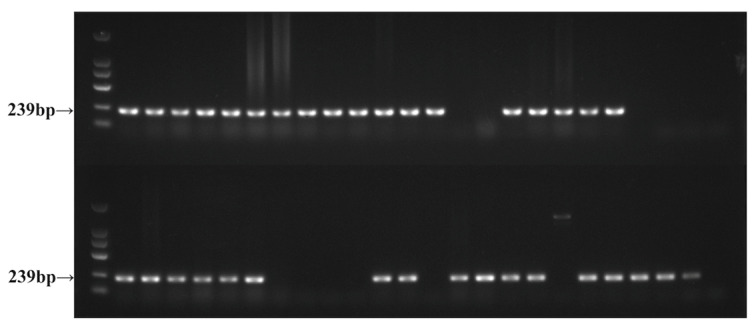
Agarose gel electrophoresis of FofraF/FofraR amplification products.

**Figure 3 ijms-26-01123-f003:**
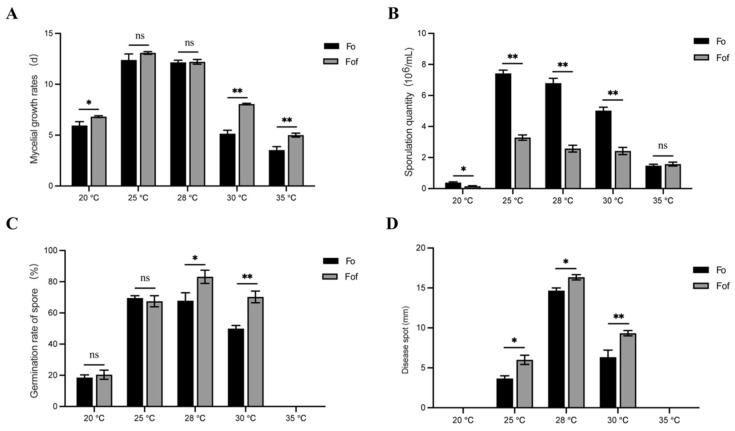
Effects of different temperatures on the (**A**) mycelial growth, (**B**) sporulation, (**C**) spore germination, and (**D**) pathogenicity of Fof and Fo. “ns” means no significant difference, * means *p* < 0.05, and ** means *p* < 0.01.

**Figure 4 ijms-26-01123-f004:**
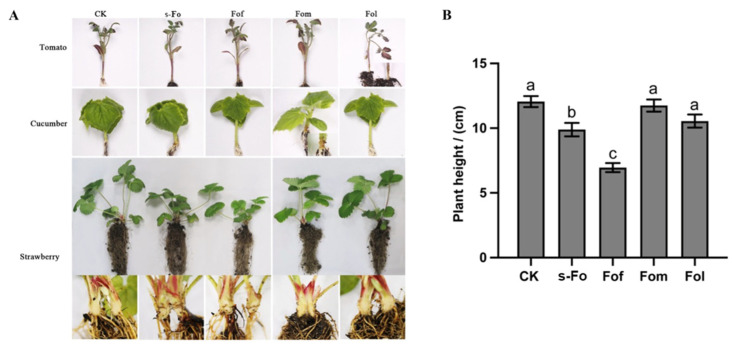
(**A**) Pathogenicity of different *Fusarium oxysporum* specialized forms to different hosts. (**B**) Effects of different *F*. *oxysporum* formae speciales on strawberry plant height. Different letters indicate a significant difference (*p* < 0.05), while the same letter indicates no significant difference. CK is treated with sterile water using the same inoculation method. s-Fo: *F. oxysporum* from strawberry; Fof: *F. oxysporum* f. sp. *fragariae*; Fol: *F. oxysporum* f. sp. *Lycopersici*; Fom: *F. oxysporum* f. sp. *Melonis*.

**Figure 5 ijms-26-01123-f005:**
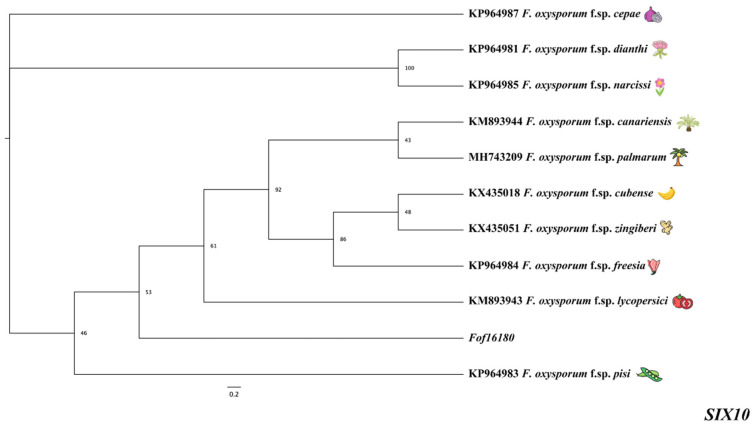
Phylogenetic tree of the Fof isolate with the *SIX10* gene of other host-specialized isolates.

**Figure 6 ijms-26-01123-f006:**
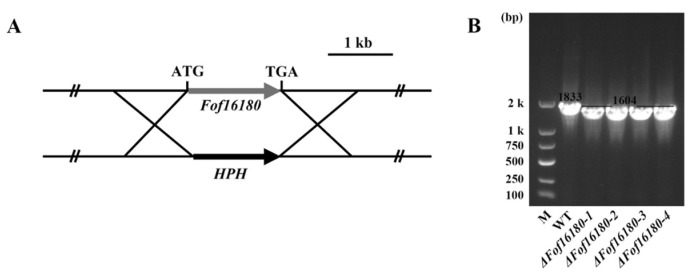
Generation and identification of *Fof16180* deletion mutants by gene replacement. (**A**) Schematic representation of the *Fof16180* replacement strategy. (**B**) PCR verification of the *Fof16180* deletion mutation.

**Figure 7 ijms-26-01123-f007:**
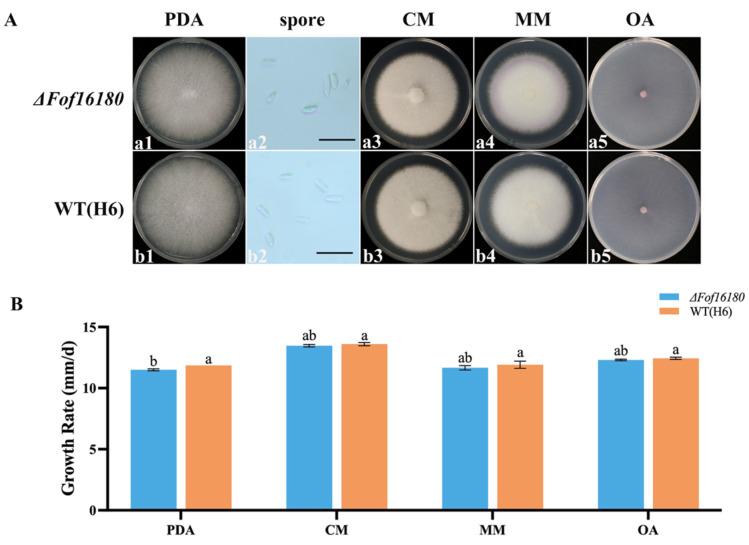
(**A**) Colony morphology of wild-type H6 and *ΔFof16180* on PDA (a1,b1), conidia (a2,b2), (scale bar: (a2,b2) = 20 µm), CM (a3,b3), MM (a4,b4), and OA (a5,b5). (**B**) Growth rate differences between H6 and *ΔFof16180* on different culture media. Bars with the same letters are not statistically different (*p* > 0.05) according to the least significant difference (LSD) test.

**Figure 8 ijms-26-01123-f008:**
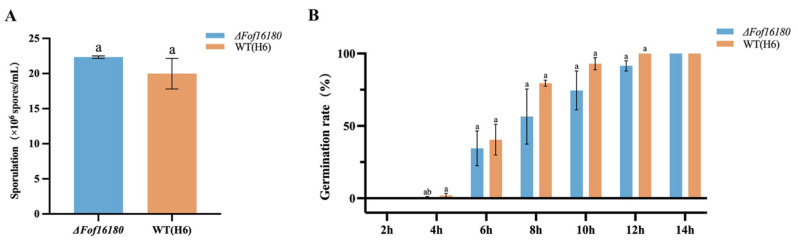
(**A**) Sporulation difference analysis of H6 and *ΔFof16180*. (**B**) Germination rate difference analysis of H6 and *ΔFof16180*. Bars with the same letters are not statistically different (*p* > 0.05) according to the least significant difference (LSD) test.

**Figure 9 ijms-26-01123-f009:**
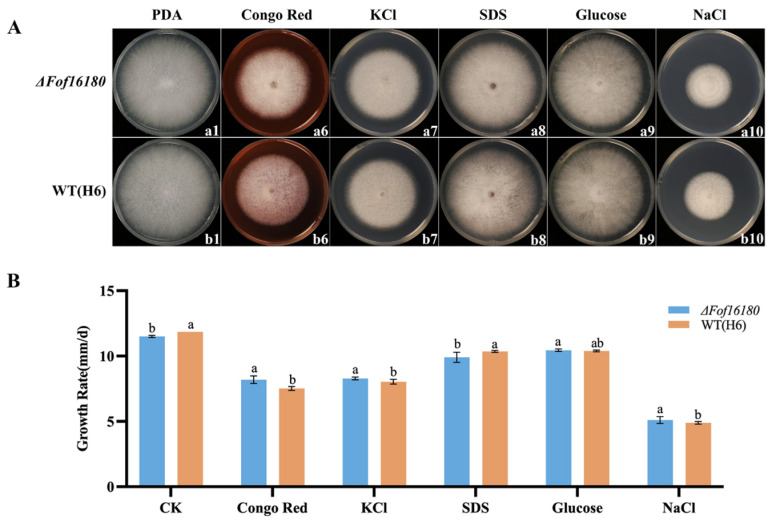
(**A**) Colony morphology of wild-type H6 and *ΔFof16180* from PDA (a1,b1), Congo red (a6,b6), KCl (a7,b7), SDS (a8,b8), glucose (a9,b9), and NaCl (a10,b10). (**B**) Growth rate differences between H6 and *ΔFof16180* on different culture media. Bars with the same letters are not statistically different (*p* > 0.05) according to the least significant difference (LSD) test.

**Figure 10 ijms-26-01123-f010:**
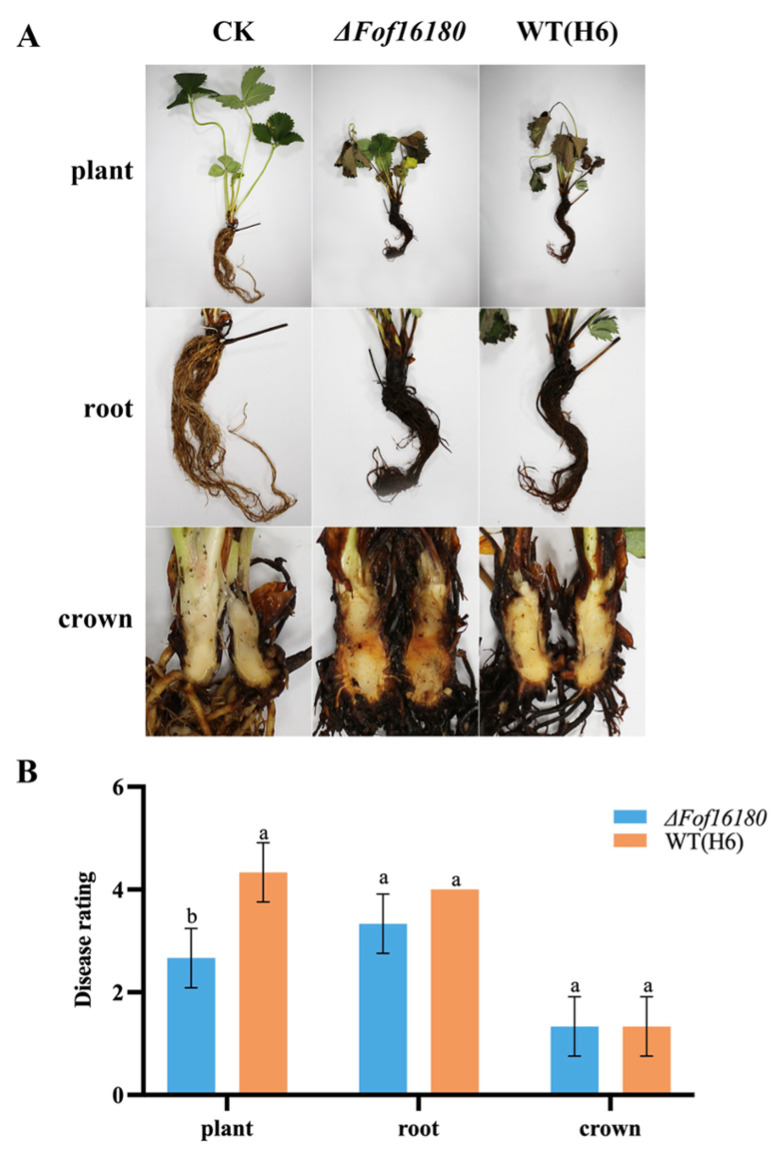
Pathogenicity of *ΔFof16180* and wild-type isolates by root inoculation. (**A**) Plant phenotype diagram. (**B**) Pathogenicity difference analysis of H6 and *ΔFof16180*. Bars with the same letters are not statistically different (*p* > 0.05) according to the least significant difference (LSD) test.

**Figure 11 ijms-26-01123-f011:**
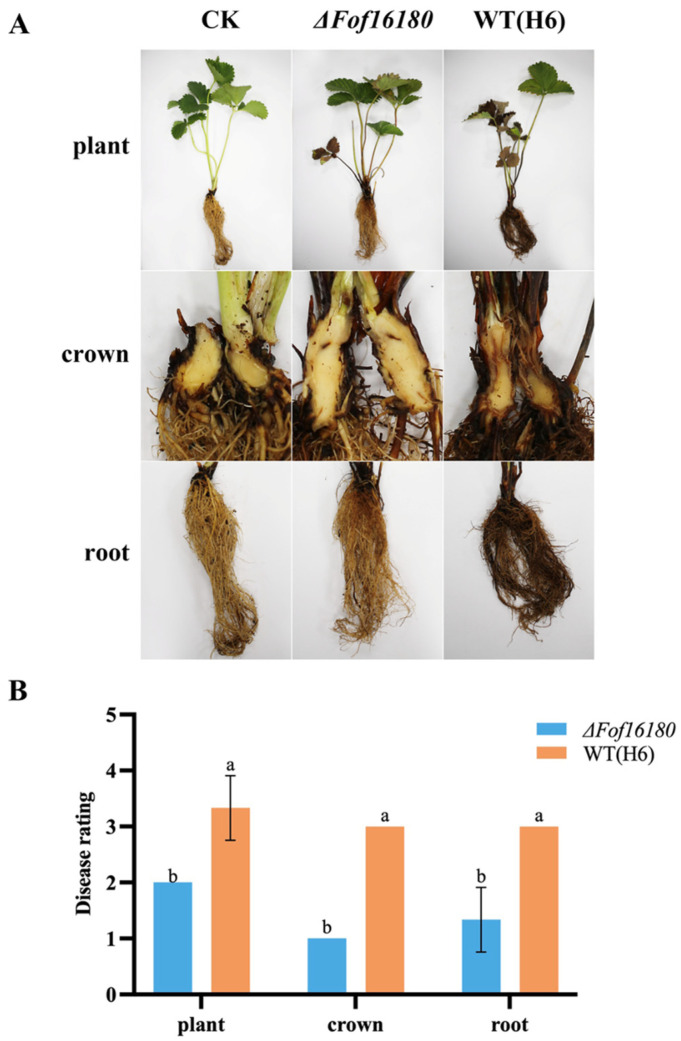
Pathogenicity of *ΔFof16180* and wild-type isolates by crown inoculation. (**A**) Plant phenotype diagram. (**B**) Pathogenicity difference analysis of H6 and *ΔFof16180*. Bars with the same letters are not statistically different (*p* > 0.05) according to the least significant difference (LSD) test.

**Table 1 ijms-26-01123-t001:** Separate isolates from various provinces.

Province	Number of Isolates	*Fusarium oxysporum*	*F. oxysporum* f. sp. *fragariae*	Proportion of Fof in This Province (%)
Anhui	8	1	7	87.50
Guizhou	5	5	0	0
Hubei	1	0	1	100
Liaoning	2	2	0	0
Yunnan	18	2	16	88.89
Zhejiang	17	3	14	82.35
Sichuan	4	3	1	25.00
Total	55	17	39	70.91

**Table 2 ijms-26-01123-t002:** Primers used in this study.

Primers	Direction	Product Length (bp)	Sequence (5′→3′)
EF1	Forward	680	ATGGGTATAAGGA(A/G) GACAAGAC
EF2	Reverse		GGA(G/A) GTACCAGT(G/A) ATCATGTT
ITS1	Forward	593	CTTGGTCATTTAGAGGAAGTAA
ITS4	Reverse		TCCTCCGCTTATTGATATGC
*Fof16180*-UP-F	Forward	1045	GGCCCAACTAAGCTTGCTTAG
*Fof16180*-UP-R	Reverse		AAAATAGGCATTGATGTGTTGACCTCCGCAAGTCGAACGAAGATAAAC
*Fof16180*-DOWN-F	Forward	978	CTCGTCCGAGGGCAAAGGAATAGAGTAGGAGCTGGACGGCGAATAATAC
*Fof16180*-DOWN-R	Reverse		CGTTGTCTATGTAGGCTGCCG
*Fof16180*-ID-F	Forward	1604	CGCAAGAGGGAATCTAGAGTC
*Fof16180*-ID-R	Reverse		CTTCGGTGACCAGCTTCGGAG
*Fof16180*-Nest-F	Forward	3089	GGCTACGCTAAGCTATTCACG
*Fof16180*-Nest-R	Reverse		GACTTTCGTCTTTGAAATAC
HPH-F	Forward	1349	GGAGGTCAACACATCAATGCCTATT
HPH-R	Reverse		CTACTCTATTCCTTTGCCCT
FofraF	Forward	239	CAGATGGGGTGCTTAAAGTT
FofraR	Reverse		ACCGCTAGGTCGTAACAAA

**Table 3 ijms-26-01123-t003:** Disease index.

Method	Root Inoculation			Crown Inoculation		
Isolate	Plant	Root	Crown	Plant	Crown	Root
*ΔFof16180*	53.33	66.67	26.67	40.00	20.00	26.67
WT (H6)	86.67	80.00	26.67	66.67	60.00	60.00

**Table 4 ijms-26-01123-t004:** Preparation methods of the media.

Media Name	Method
PDA (1 L)	Comprised 200 g of potato, 20 g of glucose, and 20 g of agar [39].
Yeast extract peptone dextrose medium (YEPD 1 L)	Comprised 10 g of peptone, 3 g of yeast extract, and 20 g of glucose. After dissolving these components, the pH was adjusted to 6.7.
Regeneration medium (RM, 1 L)	Comprised 1 g of yeast extract, 1 g of casamino acid, and 274 g of sucrose. After dissolving these components, the pH was adjusted to 7.0.
Selective regeneration medium (SRM, 1 L)	Comprised 1 g of yeast extract, 1 g of casamino acid, 342 g of sucrose, and 10 g of agarose.
Sorbitol Tris CaCl (STC, 200 mL)	Comprised 29.16 g of 0.8 M sorbitol, 1.2114 g of Tris-HCL, and 1.11 g of 50 mM CaCl. After dissolving these components, the pH was adjusted to 7.0.
Sorbitol PEG Tris CaCl (SPTC, 100 mL)	A total of 40 g of polyethylene glycol (PEG) was dissolved in 100 mL of STC.
Complete medium (CM, 1 L)	Comprised 50 mL of 20× nitrate salts, 1 g of casamino acid, 10 g of glucose, 2 g of peptone, 1 g of yeast extract, 1 mL of trace element, and 1 mL of vitamin solution. After dissolving these components in 1 L H_2_O, the pH was adjusted to 6.5.
Trace element (100 mL)	Comprised 2.2 g of ZnSO_4_·7H_2_O, 1.1 g of H_3_BO_3_, 0.5 g of MnCl_2_·4H_2_O, 0.5 g of FeSO_4_·7H_2_O, 0.17 g of CoCl_2_·6H_2_O, 0.16 g of CuSO_4_·5H_2_O, 0.15 g of Na_2_MoO_4_·5H_2_O, and 5 g of Na_4_EDTA.
Vitamin solution (100 mL)	Comprised 0.01 g of biotin, 0.01 g of pyridoxin, 0.01 g of thiamine, 0.01 g of riboflavin, 0.01 g of p-aminobenzonic acid, and 0.01 g of nicotinic acid.
Minimal medium (MM, 1 L)	Comprised 30 g of sucrose, 1 g of KH_2_PO_4_, 0.5 g of MgSO_4_·7H_2_O, 0.01 g of FeSO_4_·7H_2_O, 0.5 g of KCl, 2 g of NaNO_3_, and 200 μL of 1× trace element.
1× trace element (100 μL)	Comprised 5 g of citric acid, 5 g of ZnSO_4_·7H_2_O, 1 g of Fe (NH_4_)_2_(SO_4_)_2_·6H_2_O, and 0.25 g of CuSO_4_·5H_2_O.
Oatmeal agar (OA, 1 L)	Comprised 30 g of oatmeal and 16 g of agar.
Water agar medium (WAM, 1 L)	Comprised 16 g of agar.
0.2 g/L Congo red	A total of 0.04 g of solid Congo red was dissolved in 200 mL of PDA.
1 mol/L KCl	A total of 15.088 g of solid KCl was dissolved in 200 mL of PDA.
10% SDS	A total of 0.1 g of solid SDS was dissolved in 200 mL of PDA.
1 mol/L glucose	A total of 39.634 g of solid glucose was dissolved in 200 mL of PDA.
1 mol/L NaCl	A total of 11.688 g of solid NaCl was dissolved in 200 mL of PDA.

## Data Availability

Data are contained within the article.

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
