# Peer review of "Involvement of the SIX10 Gene in the Pathogenicity of Fusarium oxysporum Formae Speciales in Strawberries"

_ijms, 2025, doi:10.3390/ijms26031123_

Round 1
Reviewer 1 Report
Comments and Suggestions for Authors
The paper of Yang and co-authors aims to shed ligth on the role of SIX10 gene in the interaction between Fusarium oxysporum and the host strawberry. I suggest some revisions need before publication on IJMS.
Comments for authors are reported in the file attached because line numbers were not reported in the original manuscript.

Author Response
Thanks for your suggestion. We accept almost all and revised the manuscript.
Reviewer 2 Report
Comments and Suggestions for Authors
A total of 55 Fusarium oxysporum isolates obtained from strawberry plant samples collected in several Chinese provinces were characterized in the study. In the Introduction section, the Authors should also describe other strawberry diseases that pose a threat to this plant species in China. The aim of the study should be clearly stated in the Abstract and at the end of the Introduction section. The Authors used modern molecular techniques for pathogen identification, and the experiments also involved pathogen mutants from which the SIX10 gene had been removed. The results of the study are well documented and illustrated. The Discussion section should be expanded, to include e.g. a description of other F. oxysporum genes responsible for the pathogenicity of this fungal species.
The title of the manuscript is too long and should be shortened, instead of:
“Involvement of SIX10 in the Growth, Reproduction, and Pathogenicity in Strawberry Formae Speciales of Fusarium oxysporum Responsible for a Partial Pathogen Population on Strawberry Disease”
it could read:
“Involvement of the SIX10 Gene in the Pathogenicity of Fusarium oxysporum Formae Speciales in Strawberries”
The construction of phylogenetic trees raises certain doubts, and they should be modified to include a greater number of known sequences of F. oxysporum, as in the previously published article:
Hassan, O., Chang, T. Morphological and molecular characteristics of fungal species associated with crown rot of strawberry in South Korea. Mol Biol Rep 49, 51–62 (2022). https://doi.org/10.1007/s11033-021-06841-9
The research methods applied in the study are appropriate, but their description should be revised. For example, instead of describing the composition of culture media in a separate subsection (4.1. Media), it could be presented in a table or included in the subsections describing their application.
Subsection 2.3. Growth, Sporulation, and Germination Abilities of Fo and Fof at Different Temperatures
should be rewritten. The results should be described in a clear and synthetic manner, and only statistically significances should be presented.
Figure 4: the abbreviations and symbols (CK, s-Fo, Fof, Fom, Fol) should be defined.
2.6. Growth, Spore Production, and Spore Germination of ΔFof16180
The differences that are not statistically significant should not be described or discussed in the manuscript.
Author Response
A total of 55 Fusarium oxysporum isolates obtained from strawberry plant samples collected in several Chinese provinces were characterized in the study. In the Introduction section, the Authors should also describe other strawberry diseases that pose a threat to this plant species in China. The aim of the study should be clearly stated in the Abstract and at the end of the Introduction section. The Authors used modern molecular techniques for pathogen identification, and the experiments also involved pathogen mutants from which the SIX10 gene had been removed. The results of the study are well documented and illustrated. The Discussion section should be expanded, to include e.g. a description of other F. oxysporum genes responsible for the pathogenicity of this fungal species.
- The title of the manuscript is too long and should be shortened, instead of:
“Involvement of SIX10 in the Growth, Reproduction, and Pathogenicity in Strawberry Formae Speciales of Fusarium oxysporum Responsible for a Partial Pathogen Population on Strawberry Disease”
it could read:
“Involvement of the SIX10 Gene in the Pathogenicity of Fusarium oxysporum Formae Speciales in Strawberries”
Response:We accept and changed the title.
- The construction of phylogenetic trees raises certain doubts, and they should be modified to include a greater number of known sequences of oxysporum, as in the previously published article:
Hassan, O., Chang, T. Morphological and molecular characteristics of fungal species associated with crown rot of strawberry in South Korea. Mol Biol Rep 49, 51–62 (2022). https://doi.org/10.1007/s11033-021-06841-9
Response: The phylogenetic tree established by the author can already reflect the classification of strains.
3.The research methods applied in the study are appropriate, but their description should be revised. For example, instead of describing the composition of culture media in a separate subsection (4.1. Media), it could be presented in a table or included in the subsections describing their application.
Response: Already modified with the Table 4.
4.Subsection 2.3. Growth, Sporulation, and Germination Abilities of Fo and Fof at Different Temperatures
should be rewritten. The results should be described in a clear and synthetic manner, and only statistically significances should be presented.
Response: it was rewritten as: Fof and Fo grew under five temperature conditions, with Fof showing stronger adaptability to high temperatures (Figure 3A). At 25 and 28°C, there were no significant differences in the growth rate (p > 0.05). At 20°C, the mycelial growth rate of Fof was significantly higher than that of Fo (p < 0.05). At 30 and 35°C, the mycelial growth rate of Fof was significantly higher than that of Fo (p < 0.01).
Fof and Fo both produced spores under five temperature conditions, with Fo having significantly higher spore production than Fof (Figure 3B). The spore production of Fo was significantly different (p < 0.05) at 20, 25, 28, and 30°C. The spore production of Fo was significantly higher than that of Fof (p < 0.05), but there was no significant difference in spore production between Fof and Fo at 35°C (p > 0.05).
At 35°C, Fof and Fo spores did not germinate. At 20 and 25°C, there were no significant differences in the spore germination rate between Fof and Fo (p > 0.05). At 28°C, the spore germination rate of Fof was significantly higher than that of Fo (p < 0.05). At 30°C, the spore germination rate of Fof was significantly higher than that of Fo (p < 0.01) (Figure 3C).
The pathogenicity of Fof was significantly higher than that of Fo. Fo and Fof were not pathogenic to strawberry plants at 20 or 35°C, but they had the strongest pathogenicity at 28°C (Figure 3D).
- Figure 4: the abbreviations and symbols (CK, s-Fo, Fof, Fom, Fol) should be defined.
Response: Their are been defined as: CK is treated with sterile water using the same inoculation method.s-Fo:F. oxysporum from strawberry; Fof :F. oxysporum f. sp. fragariae; Fol : F. oxysporum f. sp. Lycopersici; Fom : F. oxysporum f. sp. Melonis.
Figure 4: It has been supplemented in the textual description section of 2.4.
- 6. Growth, Spore Production, and Spore Germination of ΔFof16180
The differences that are not statistically significant should not be described or discussed in the manuscript.
Response: Already modified.